# A Miniaturized Wideband Bandpass Filter Using Quarter-Wavelength Stepped-Impedance Resonators

**Liqin Liu [1], Ping Zhang [1], Min-Hang Weng [1]**  **, Chin-Yi Tsai [2] and Ru-Yuan Yang [2,\*]**

[1]  School of Information Engineering, Putian University, Putian 351100, Fujian, China; llqsmile006@gmail.com (L.L.); zhangpingdudu@gmail.com (P.Z.); hcwweng@gmail.com (M.-H.W.)
[2]  Graduate Institute of Materials Engineering, National Pingtung University of Science and Technology, Pingtung County 912, Taiwan; tylertsaiji@gmail.com
\*  Correspondence: ryyang@mail.npust.edu.tw; Tel.: +886-8-7703202

**Abstract:** In this paper, we present a simple method to design a miniaturized wideband bandpass filter with suppression of the third harmonic, using only two quarter-wavelength stepped-impedance resonators (SIRs). The resonant modes of the quarter-wavelength SIR, depending on the impedance ratio (K) and electrical length ratio ($\alpha$), are discussed first. As to setting the resonant frequency of the SIR for the lower band edge of the required band, the size parameters of two quarter-wavelength SIRs can be determined by selecting the desired impedance ratio (K) and length ratio ($\alpha$). By using the opposite directional arrangement of two SIRs with direct taped input/output ports, the wideband response can be formed. A filter example is shown in this study to address this simple design procedure. The measured results of the fabricated filter have a wide passband response from 3.3 to 5.8 GHz, with an insertion loss of 1.5 dB, a return loss of 20 dB, an extended bandwidth ration of 55%, a low-average group delay of less than 0.75 ns, and a stopband from 6 to 12 GHz, with an attenuation level of 20 dB. Due to the similar 0° feeding, a transmission zero at 8.3 GHz appears near the band edge; thus, improving the band selectivity. The proposed filter can have a very simple structure and a miniature size. Simulated results and measured results are in good agreement.

**Keywords:** wideband; bandpass filter; quarter wavelength; stepped-impedance resonator (SIR)

## 1. Introduction

In the last 20 years, communications systems have developed rapidly. The bandpass filter (BPF) used in the radio-frequency (RF) front end is an important device for selecting the desired signals for the use of the communications system [1]. Typically, the requirements of the filter comprise low passband loss, sharp band selectivity, spurious suppression, a compact size, and a low cost. The wideband system has been rapidly expanding ever since in 2002 the U.S. Federal Communications Commission (FCC) approved the unlicensed applications of ultra-wideband (UWB) with frequency ranging from 3.1 to 10.6 GHz for several uses, such as hand-held and indoor systems [2]. The direct sequence ultra-wideband (DS-UWB) specifications for wireless personal area networks (WPANs) is further divided into a low band of 3.1–5.1 GHz and a high band of 6.2–9.7 GHz, to avoid the frequent use of IEEE 802.11a wireless local area networks (WLANs) at 5–6 GH. As one of the important component blocks, some wideband BPFs were developed to obtain the desired fractional bandwidth (FBW) [3–15].

In Chang's work [3], a wide BPF adopted a combination of serial and shunt microstrip units to have an FBW of 54%. However, the drawbacks of this filter are the large size and complex structure. In Hung's work [4], a wideband filter using parallel coupled lines was reported to obtain an FBW of 80%, with a complex image impedance design method. Various types of multiple-mode resonators (MMRs) were used to obtain the wideband filter. In Zhu's work [5,6], the resonant modes of the MMR

were analyzed first, and then controlled to be coupled together to obtain the desired bandwidth. In Killamsetty's work [7], a short-circuited loaded triangular stub resonator was used to obtain a wideband BPF, but the structure was still complex. In Wang's work [8] and Ye's work [9], ultra-wideband (UWB) BPFs were designed by semi-lumped or hybrid microstrip/coplanar waveguide (CPW) structures. In Chang's work [10], a stepped-impedance resonator (SIR) was developed to realize a wideband response, which showed a high band selectivity. In Choudhary's work [11], split circular rings and a rectangular stub were used to design a via-less metamaterial wideband BPF, but the bandwidth was insufficiently large. In Ji's work [12], a multilayer structure was used to achieve a wideband BPF. In Li's work [13], series and shunt resonators were coupled together to directly realize a wideband BPF. In Li's work [14], a single wavelength ring SIR was used to implement a wide-frequency band. In Gao's work [15], a combination of open/shorted stubs was used to obtain a wideband response with an improved upper-stopband. In Hameed's work [16], multiple-mode split-ring resonators in a waveguide cavity were designed to obtain a wideband BPF. However, the above design procedure and the device structure are complex.

In this study, a simple method is reported to design a wideband BPF with an FBW greater than 50%, and with suppression of the third harmonic. Only two quarter-wavelength SIRs are needed in this design. The design concept and procedure are described in this study. The designed filter can simultaneously have a simple structure and miniature size. To prove the design concept, a filter example is presented. The measured results of the fabricated filter match with the simulated results, showing that the filter can be suitable for a practical RF system.

## 2. Design Procedure

Figure 1 displays the structure of the proposed wide BPF. The basic element of the filter is two microstrip quarter-wavelength SIRs. $(L_1, L_2)$ and $(W_1, W_2)$ are the physical lengths and widths of the high impedance section and low impedance section of SIR 1, respectively. $(L_3, L_4)$ and $(W_3, W_4)$ are the physical lengths and widths of the high impedance section and low impedance section of SIR 2, respectively. Gap (g) is the spacing between SIR 1 and SIR 2. Expression (p) is the physical length from the input/output ports to the short ends of SIR 1 and SIR 2. For this design, a low-cost FR4 substrate is used, having a dielectric constant $(\varepsilon_r)$ of 4.4, a loss tangent $(\tan\delta)$ of 0.02, and a thickness of 1.6 mm.

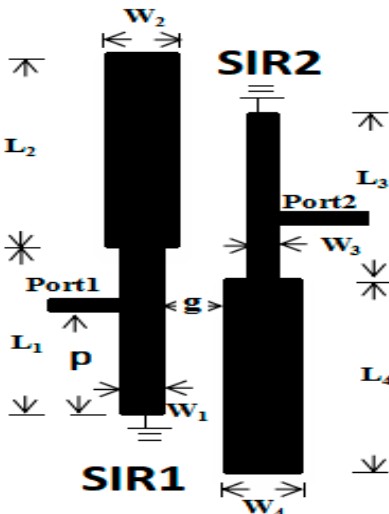

**Figure 1.** The layout of the proposed wideband bandpass filter (BPF).

*Analysis of Quarter-Wavelength Stepped-Impedance Resonator*

Figure 2a displays the construction of the quarter-wavelength SIR. The SIR is formed by an impedance unit $(Z_1)$ with an electrical length $(\theta_1)$, and an impedance unit $(Z_2)$ with an electrical length $(\theta_2)$. The impedance ratio of this quarter-wavelength SIR is defined as $K = Z_2/Z_1$ and the electrical

length ratio is defined as $\alpha = \theta_2/(\theta_1+\theta_2) = \theta_2/\theta_t$. The characteristics of the conventional SIR are able to control the higher-order resonance modes, closer or far away, efficiently, with a different impedance ratio (K) and electrical length ratio ($\alpha$). The input impedance ($Z_{in}$) of the quarter-wavelength SIR is derived as [17]:

$$Z_i = jZ_2 \frac{Z_1 \tan \theta_1 + Z_2 \tan \theta_2}{Z_2 - Z_1 \tan \theta_1 \tan \theta_2}. \tag{1}$$

The resonant condition can be obtained when $Y_{in} = 0$ as follows [1]:

$$\tan \theta_1 \tan \theta_2 = K \tag{2}$$

To reduce the design parameters, $\theta_1$ and $\theta_2$ are derived by $\alpha$ and $\theta_t$ as follows:

$$\theta_1 = (1 - \alpha) \cdot \theta_t \tag{3}$$

$$\theta_2 = \alpha \cdot \theta_t \tag{4}$$

Thus, the resonant condition controlled by the impedance ratio (K) and the electric length ratio ($\alpha$) is expressed as:

$$K = \tan[(1 - \alpha) \cdot \theta_t] \cdot \tan(\alpha \cdot \theta_t). \tag{5}$$

Figure 2b illustrates the resonant condition curve of the quarter-wavelength SIR. As shown in Figure 2b, the total electrical length becomes shorter and longer when the impedance ratio (K) is smaller and larger than 1, respectively. In other words, for the same resonator size, the frequency of the quarter-wavelength resonator shifts lower and higher when the impedance ratio (K) is smaller and larger than 1, respectively [17].

Figure 3 illustrates (a) the structure of the quarter-wavelength SIR with a special case of K = 1 and (b) the simulated frequency responses with different feeding positions of the input and output ports. For the special case of K = 1, the quarter-wavelength SIR is seen as the conventional quarter-wavelength uniform impedance resonator (UIR). Based on the coupled line theory, the conventional quarter-wavelength UIR with opposite directions would have a passband response. The bandwidth of the passband can be controlled by the impedance and gap of the coupled lines, namely the even mode impedance $Z_{even}$ and odd mode impedance $Z_{odd}$ of the coupled lines [18]. Moreover, there are many harmonics that appeared in the higher frequency. The frequency of the first harmonic is 3 times the fundamental mode, as shown in Figure 2c. Moreover, it is known that no matter what value $\alpha$ is, the fundamental resonant mode is kept and resonant, as $(\theta_1+\theta_2) = \theta_t$ is equal to the quarter-wavelength, as shown in Figure 2b.

It is known that the feeding positions (t) of the input and output (I/O) ports on the resonators affect the filter performances. Figure 3b shows the simulated filter responses of the quarter-wavelength UIR with different feeding positions (t) of the input and output (I/O) ports, where t is the physical length from the center of the quarter-wavelength UIR to the short end. The simulation was done by using the full-wave electro-magnetic simulator IE3D [19]. In this simulation, $L_1+L_2$ and $L_3+L_4$ were both selected and kept as 11 mm to be the quarter-wavelength at 4 GHz. The impedance of the UIR was 100 $\Omega$ and the gap (g) was 2 mm. It was clearly found that as the feeding positions of the I/O ports are separated away, there were two modes that existed in the resonator. As shown in Figure 3b,c, when t is increased from 0 mm to 1 mm, a wideband response is formed and two separated modes appear. However, as t is further increased from 1 mm to 4 mm, the band response is degraded since the modes are separated and not coupled together. The first resonant mode at lower frequency is the fundamental mode of the quarter-wavelength UIR. Since the open stub can also be seen as the quarter-wavelength resonator, the other mode at higher frequency is resonant at the length from the I/O ports to the open end ($t + L_2$ ($L_4$)). Namely, the frequency of the other mode can be roughly estimated by seeing that the length ($t + L_2$ ($L_4$)) is a quarter-wavelength. Therefore, by carefully selecting the feeding positions of the I/O ports, the desired passband response can be formed. At t = 1 mm, the

bandwidth of the passband is from 4 to 6.2 GHz, namely, the FBW is 43%. However, it is also known that the next harmonic also appeared at around 12 GHz, which is 3 times the fundamental frequency of the quarter-wavelength resonator.

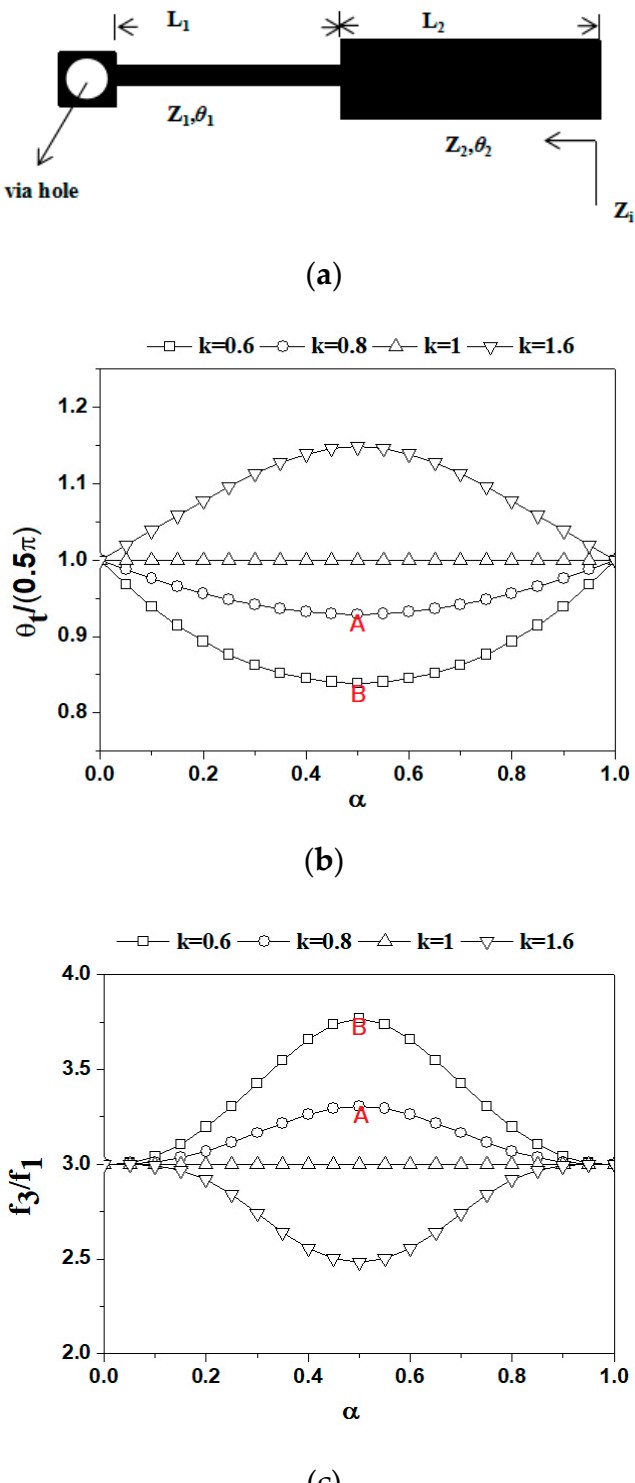

**Figure 2.** (**a**) The construction and (**b**) resonant condition curve (**c**) $f_3/f_1$ of the quarter-wavelength stepped-impedance resonator (SIR).

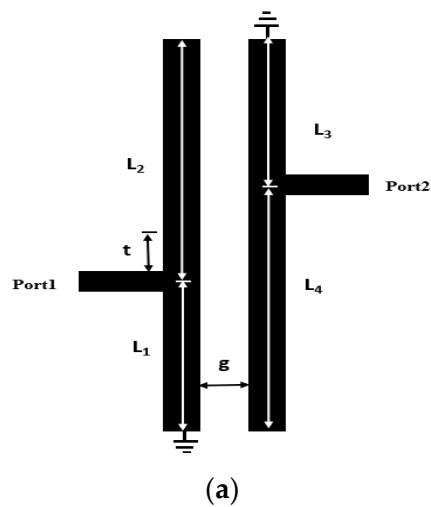

(**a**)

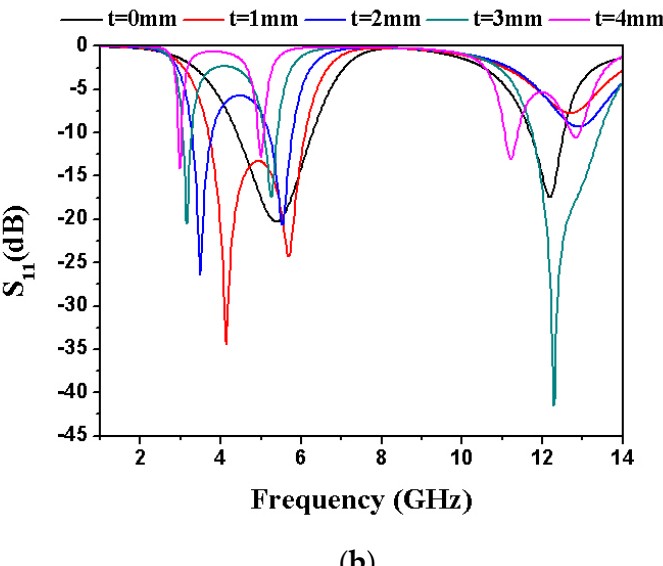

(**b**)

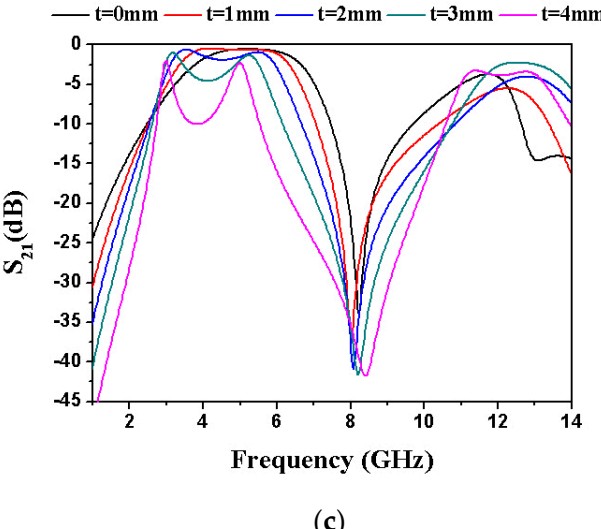

(**c**)

**Figure 3.** (**a**) the structure of the quarter-wavelength SIR with special case of K = 1 and (**b**) the simulated frequency responses with different feeding positions (**c**) of the input and output ports.

Figure 4 shows the current distribution of the quarter-wavelength UIR at 3.2, 3.6, 5.2, and 5.6 GHz. In this simulation, t was 1 mm, and a wideband response with an FBW of 43% was obtained, as shown in Figure 3c. Typically, the current distribution of the filter at the resonant frequency is used to show the locations of the maximum and minimum electromagnetic energy, namely, to know where resonance occurs in the structure. This resonant mode can be then excited by providing the suitable input and output terminals in the resonant excitation location. The resonant mode can be suppressed if a dispersing device is used in the resonant excitation location to avoid the resonant mode. As shown in Figure 4, the resonant energy at lower frequency (see 3.2 GHz and 3.6 GHz) is distributed over the UIR, and the resonant energy at higher frequency (see 5.2 GHz and 5.6 GHz) is distributed mostly on the area from the I/O ports to the open end (t+ $L_2$ ($L_4$)).

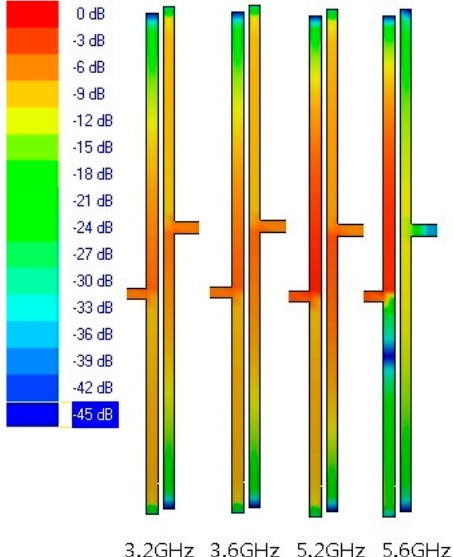

**Figure 4.** Current distribution of the quarter-wavelength uniform impedance resonator (UIR) at 3.2, 3.6, 5.2, and 5.6 GHz.

Based on the above discussion, in this study, to further extend the bandwidth of the wideband filter, the quarter-wavelength UIR (K = 1) was replaced with the quarter-wavelength SIR. The filter example is designed around 4.2 GHz. The resonant frequencies of the quarter-wavelength SIR were selected, first based on Figure 2b,c, and then on a wide bandpass response that can easily be achieved by coupling the resonant modes with a careful arrangement of the I/O ports. As mapping spot A and spot B in Figure 2b for SIR 1 and SIR 2, respectively, (K = 0.8, $\alpha$ = 0.5) and (K = 0.6, $\alpha$ = 0.5) were selected in this design for miniaturizing the filter size and providing two lower resonant modes at 3.9 GHz and 3.5 GHz, respectively. Therefore, for the quarter-wavelength SIR 1, the physical width and length are 0.35 mm ($W_1$) and 5 mm ($L_1$), and 1.1 mm ($W_2$) and 5 mm ($L_2$) for the high impedance section ($Z_1$ = 100 $\Omega$), and the low impedance section ($Z_2$ = 60 $\Omega$), respectively. For the quarter-wavelength SIR 2, the physical width and length are 0.35 mm ($W_3$) and 5 mm ($L_3$), and 0.6 mm ($W_4$) and 5 mm ($L_4$) for the high impedance section ($Z_1$ = 100 $\Omega$) and the low impedance section ($Z_2$ = 80 $\Omega$), respectively. The reason for using two different SIRs is to extend the bandwidth of the designed filter. Moreover, as shown in Figure 2c, the frequency of the third harmonic mode ($f_3$) over the frequency of fundamental mode ($f_1$), namely $f_3/f_1$, is moved to higher values of 3.3 and 3.7 for SIR 1 and SIR 2, as mapping the spot A and spot B in Figure 2c, respectively. Thus, the third harmonics of this filter are suppressed, since the two third harmonics of the two different SIRs (SIR1 and SIR2) are different, and cannot be coupled.

After determining the two quarter-wavelength SIRs, the input/output (I/O) ports were directly taped onto the quarter-wavelength SIRs. To achieve good external quality, and a transmission zero, a similar 0° feed structure of the I/O ports was directly taped to the two quarter-wavelength SIRs

at p = 3 mm [1]. As discussed above, in this wide passband, other higher resonant modes are also excited. When p = 3 mm, the two quarter-wavelength SIRs have total physical lengths of 7 mm from the feeding position of the I/O ports to the open ends, and would provide two other modes at 5.1 GHz and 5.6 GHz, respectively, as also mapping into Figure 2b.

Figure 5 shows the simulated filter responses when varying the coupling gap (g) between two quarter-wavelength SIRs. It was found that as the g value decreases from 0.5 to 0.05 mm, the passband became wider and the insertion loss became lower. Because the minimum carving size of the carving machine is 0.15 mm, the coupling gap, g = 0.15 mm, was used to obtain the maximum passband with a low insertion loss of 1.2 dB, a return loss of 20 dB, and a 3 dB FBW of 55% (from 3.3 to 5.8 GHz). Moreover, by taking advantage of the direct 0° feed structure of the I/O ports, a transmission zero at 8.3 GHz was generated near the passband edge in the filter response [20]. In addition, it was clearly observed that the third harmonic of the quarter-wavelength UIR near 12 GHz was suppressed; thus, a stopband with an attenuation of 15 dB from 7.0 to 12 GHz was obtained.

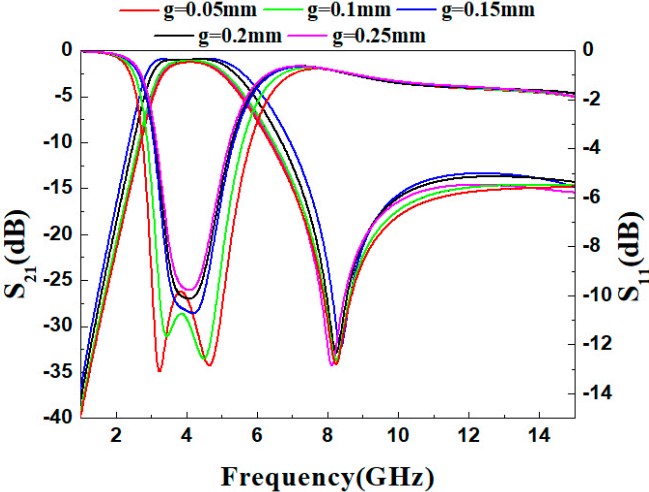

**Figure 5.** Filter responses when varying the coupling gap (g) between two quarter-wavelength SIRs.

Figure 6 further shows the current distribution of the proposed wideband filter at the frequencies of 3.5, 4.0, 5.5, and 6 GHz. As shown in Figure 6, the resonant energy at lower frequency (see 3.5 GHz and 4.0 GHz) is distributed over the SIR, and the resonant energy at higher frequency (see 5.5 GHz and 6.0 GHz) is distributed mostly on the areas from the I/O ports to the open end, thus verifying the design concept.

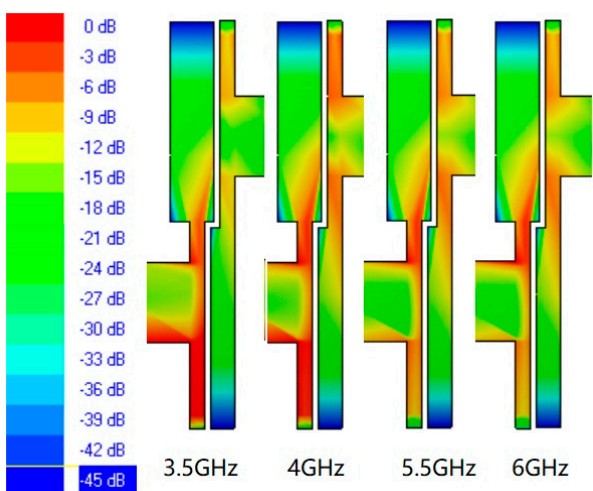

**Figure 6.** Current distribution of the proposed wideband filter at lower and higher frequencies.

With this simple method and design concept, the resonant frequencies of the quarter- wavelength SIR are selected first, and then a wide bandpass response can be easily achieved by coupling the resonant modes with a careful arrangement of the I/O ports.

## 3. Experimental Results

The filter sample was fabricated using conventional printing circuit board technology. Figure 7a shows a photograph of the fabricated sample. The whole size of the fabricated filter is 12 mm X 4 mm, i.e., approximately 0.3 λg by 0.1 λg, where λg is the guided wavelength at the center frequency. Measurement was processed by an HP8722ES network analyzer. Before measurement, two coaxial cables of the network analyzer, which were connected to the I/O ports of the fabricated filter sample, were carefully calibrated by using short-open-load-through calibration. Steps were carefully processed to make sure that the $S_{21}$ was close to zero when the two coaxial cables were connected to the load-through device. The measured results shown in Figure 7b exhibit a center frequency of 4.2 GHz with a low insertion loss of 1.2 dB over the passband, a return loss greater than 15 dB, a 3 dB FBW of 55% (from 3.3 to 5.8 GHz), and a stopband with an attenuation of 15 dB, from 7.5 to 12 GHz. Moreover, the transmission zero at 8.3 GHz was clearly obtained because of the use of a 0° feeding structure [20]; thus, a good band selectivity was also achieved. The group delay was obtained by taking the derivative of the phase. Figure 7c shows that the average calculated group delay of the fabricated filter is less than 0.75 ns over the whole passband. As compared to other works with a group delay, this group delay of this design is acceptable. The simulated results and the measured results are mostly in agreement, with a slight mismatch in the high band edge of the passband. This mismatch may have been due to the fact that the electromagnetic phenomenon of the solder at the short-circuited end was not well-considered, or the amount of solder at the short-circuited end was not appropriate.

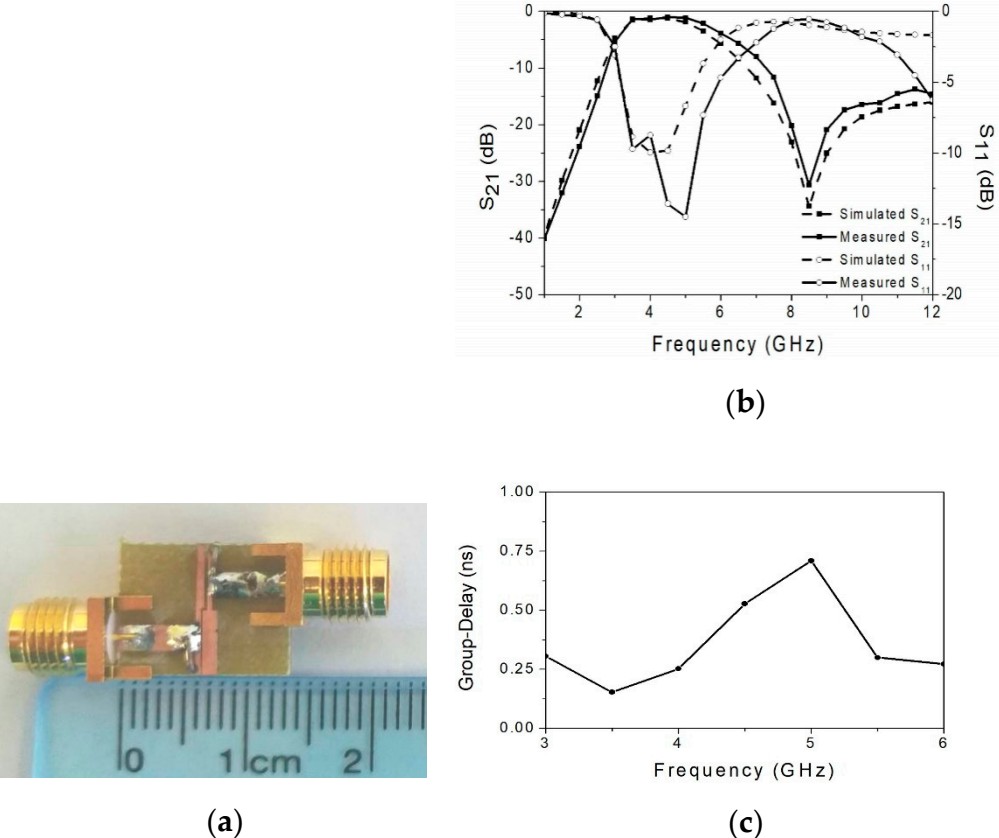

**Figure 7.** (**a**) A photograph of the fabricated sample, (**b**) measured filter responses, and (**c**) the calculated group delay of the fabricated filter.

Table 1 compares this design to some reported works. The designed filter shows acceptable filter performance when compared to other filters. In addition, this design shows a simple configuration and a miniaturized size. The filter example is designed at 4.2 GHz with a 3 dB fractional bandwidth of 55%, which can meet the low band of 3.1–5.1 GHz of the DS-UWB specification with a slight tuning of the passband. Moreover, the filter design procedure is flexible and can be designed at other frequency ranges when using the desired substrate. Therefore, because of its simple topology, miniaturized and compact size, and good performance, the designed filter is very useful for modern wideband wireless communication systems.

**Table 1.** Comparison of filter performances of the proposed filter with the previous published works.

|  | Ref. [11] | Ref. [12] | Ref. [13] | Ref. [14] | Ref. [15] | This Work |
|---|---|---|---|---|---|---|
| Center frequency (GHz) | 2.3 | 3 | 2 | 2.3 | 1 | 4.2 |
| $|S_{11}|$ (dB) | 13 | 11.7 | 20 | >13 | 15 | 15 |
| $|S_{21}|$ (dB) | 0.35 | 2.1 | 0.57 | 0.35 | 1 | 1.2 |
| 3 dB FBW (%) | 80 | 107 | 100 | 80 | 123 | 55 |
| Circuit Size ($\lambda g \times \lambda g$) | $0.12 \times 0.22$ | $0.89 \times 0.46$ | No description | $0.53 \times 0.43$ | $0.17 \times 0.14$ | $0.3 \times 0.1$ |
| Wide stopband | No | No | Yes | No | Yes | Yes |
| Defected ground | No | Yes | No | No | No | No |

## 4. Conclusions

This paper presented a simple method to design a miniaturized wideband bandpass filter with suppression of the third harmonic. The filter is formed by simply using only two quarter-wavelength stepped-impedance resonators (SIRs). For the wideband design, the resonant modes of the two quarter-wavelength SIRs are chosen first by selecting the impedance ratio (K), and length ratio ($\alpha$), to the lower band edge of the designed wideband. The I/O ports are then directly taped using a similar 0° feeding structure at the desired position of the two opposite direction quarter-wavelength SIRs. With a suitable arrangement of the two SIRs, the wideband response can be formed. A designed filter with a 3 dB fractional bandwidth of 55% was presented and fabricated in this study to verify the design concept. The measured results have a center frequency of 4.2 GHz with a low insertion loss of 1.5 dB, a return loss greater than 15 dB, and a stopband from 7.5 to 12 GHz with an attenuation of 15 dB. A transmission zero appears at 8.3 GHz to obtain an acceptable band selectivity. The average group delay is as low as it is around 0.1–0.75 ns. Moreover, the filter has a miniature size due to this simple design topology. The simulated results and the measured results are in good agreement. Based on this design concept, further works will be performed on the design of the dual wideband filter and diplexer in a miniature size.

**Author Contributions:** Conceptualization, M.-H.W.; methodology, L.L. and P.Z.; software, L.L.; validation, L.L. and P.Z.; formal analysis, L.L. and P.Z.; investigation, M.-H.W.; resources, R.-Y.Y.; data curation, C.-Y.T.; writing—original draft preparation, L.L. and M.-H.W.; writing—review and editing, L.L. and R.-Y.Y.; visualization, L.L. and P.Z.; supervision, R.-Y.Y.; project administration, R.-Y.Y.; funding acquisition, M.-H.W.

**Funding:** This work was supported by the Putian University's Initiation Fee Project for Importing Talents for Scientific Research (2019001) and (2019003).

**Acknowledgments:** The authors acknowledge Hong-Zheng Lai and Shih-Kun Liu for the help with sample measurement.

**Conflicts of Interest:** The authors declare no conflict of interest.

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
