# Peer review of "A Miniaturized Wideband Bandpass Filter Using Quarter-Wavelength Stepped-Impedance Resonators"

_electronics, doi:10.3390/electronics8121540_

Round 1

Reviewer 1 Report

The manuscript "A Miniaturized Wideband Bandpass Filter Using Quarter Wavelength Stepped-Impedance Resonators" doesn't appear to be a properly written journal paper, but more like a project report. The concept of using step impedance resonators to construct a bandpass filter around a few GHz could be of interest. If this manuscript were to be accepted, significant revision is necessary. 

I feel that many places in the work do not convey validated reasonings. Two major flaws in the article lie in (1) the lack of comparison with other techniques in bandwidth comparisons; and (2) the lack of illustrating losses in the system. Countless designs can be a bandpass filters and not all of them carry non-trivial values. Point (1) is necessary because advantages of the design must be shown to prove such point, it can be side by side comparison of a commercial one, or proof through design principle that the authors' approach is superior. Point (2) is also necessary, as I only see the amplitude attenuation chart but the phase distortion of a filter is of critical importance. It is hard to believe that the show of group delay will qualify the design as a good one, and he reviewer found no insightful discussion to qualify that. It is therefore required for both analysis and demonstration to show the merit of the design.

Other issues are listed below:

A. Do not use In [Ref] for statement in the introduction. Instead, put down the names of the major contributors, like in "someone's work on ..." 

B. In figures 1,3, the two devices have a spacing but such spacing disappear in figure 4. Please explain the inconsistency here.

C. Figure 5 should have different color or makers to make each curve discernible.

D.  Adding a section to describe the fitness or application context of the design will be helpful. Right now I feel such design an isolated subject from the major trend in the communication work.

Overall, I feel this manuscript should be significantly revised and re-reviewed before a formal decision is made.

Author Response

Answers for reviewer’s Comment

The comments from the reviewer are very constructive to our paper. The paper is revised in accordance with the reviewer’s comments and has marked the corrections using yellow color in the revised word file. The attached are the revision and the reply letter for the comments. There are listed as follows:

Answers for Reviewer 1's Comments

The manuscript "A Miniaturized Wideband Bandpass Filter Using Quarter Wavelength Stepped-Impedance Resonators" doesn't appear to be a properly written journal paper, but more like a project report. The concept of using step impedance resonators to construct a bandpass filter around a few GHz could be of interest. If this manuscript were to be accepted, significant revision is necessary. 

I feel that many places in the work do not convey validated reasonings. Two major flaws in the article lie in (1) the lack of comparison with other techniques in bandwidth comparisons; and (2) the lack of illustrating losses in the system. Countless designs can be a bandpass filters and not all of them carry non-trivial values.

Q1: Point (1) is necessary because advantages of the design must be shown to prove such point, it can be side by side comparison of a commercial one, or proof through design principle that the authors' approach is superior.

Reply:

The comment is well received. In this revised version, we have re-designed and re-fabricated the filter with a passband from 3.3-5.8 GHz to meet the useful communication system. We have also added many descriptions to address the design concepts and related theory.

New Figure 3 and 4 are added in the revised version to show our novel design concept. By using only two quarter wavelength SIRs, the advantages of the design include a simple structure, a miniaturized size and suppression of the third harmonic. We have added a comparison table in the revised version to compare the proposed design with others, to show the advantages of this design.

We have also a description with yellow mark to show the advantage of this simple design: Table 1 compares this design to some reported works. The designed filter shows acceptable filter performance when comparing to other filters. In addition, this design shows a simple configuration and a miniaturized size. The filter example is designed at 4.2 GHz with a 3-dB fractional bandwidth of 55 %, which can meet the low band of 3.1-5.1 GHz of the DS-UWB specification with a slight tuning of the passband.

Table 1. Comparison of filter performances of the proposed filter with the previous published works.

Ref. [11]

Ref.[12]

Ref.[13]

Ref. [14]

Ref. [15]

This work

Center frequency (GHz)

2.3

3

2

2.3

1

4.2

|S11| (dB)

13

11.7

20

>13

15

15

|S21| (dB)

0.35

2.1

0.57

0.35

1

1.2

3-dB FBW (%)

80

107

100

80

123

55

Circuit Size (λg×λg)

0.12 ×

0.22

0.89 × 0.46

No description

0.53 × 0.43

0.17× 0.14

0.3 ×

0.1

Wide stopband

No

No

Yes

No

Yes

Yes

Defected ground

No

Yes

No

No

No

No

Q2: Point (2) is also necessary, as I only see the amplitude attenuation chart but the phase distortion of a filter is of critical importance. It is hard to believe that the show of group delay will qualify the design as a good one, and the reviewer found no insightful discussion to qualify that. It is therefore required for both analysis and demonstration to show the merit of the design.

Reply:

The comment is well received. The group delay is obtained by taking the derivative of the phase and Figure. 7(c) shows that the average calculated group delay of the fabricated filter is less 0.75ns over the whole passband. As compared to other works with group delays which are located between 0.1~1.5 ns, this group delay of this design is acceptable. In this revised version, we have re-designed and re-fabricated the filter with a passband from 3.3-5.8 GHz to meet the useful communication system. We have also added many descriptions to address the design concepts and related theory. We have carefully presented our analysis and demonstration to show the merit of the design. The advantages of the design include a simple structure, a miniaturized and compact size and suppression of the third harmonic. We have added a comparison table in the revised version to compare the proposed design with others, to show the advantages of this design.

Other issues are listed below:

Q3: A. Do not use In [Ref] for statement in the introduction. Instead, put down the names of the major contributors, like in "someone's work on ..." 

Reply:

The comment is well received. We have amended all the statement of previous works in the introduction by putting down the names of the major contributors. All the corrections have been marked with yellow color in the revised word file.

Q4: B. In figures 1,3, the two devices have a spacing but such spacing disappear in figure 4. Please explain the inconsistency here.

Reply:

The comment is well received. Since the spacing between two SIRs is only 0.15mm, which is very small, as compared to other structure parameters of two SIRs. Thus, the spacing in figure 4 seems disappear. However, the spacing of 0.15 mm is actually existed in the simulated current distribution. We have carefully amended this misunderstanding by enlarging the figure 4 (current distribution, now is figure 6)

Figure 6

Q5: C. Figure 5 should have different color or makers to make each curve discernible.

Reply:

The comment is well received. We have amended the plot by using different colors to make each curve discernible in the revised version, as below. We have also a description with yellow mark to show the advantage of this simple design: In addition, it is clearly observed the third harmonic of the quarter wavelength UIR near 12 GHz is suppressed, thus a stopband with an attenuation of 15 dB from 7.0 GHz to 12 GHz is obtained.

Figure 5 (Figure 7)

Q6: D.  Adding a section to describe the fitness or application context of the design will be helpful. Right now I feel such design an isolated subject from the major trend in the communication work.

Overall, I feel this manuscript should be significantly revised and re-reviewed before a formal decision is made.

Reply:

The comment is well received. We have re-designed the filter example to have an application of lower band (3.1 to 5.1 GHz,) of ultra-wideband (UWB) range from 3.1 to 10.6 GHz, established by the U. S. Federal Communications Commission (FCC) since February 2002, which can be used to apply to imaging systems, vehicular radar systems, communication and measurement systems, etc. Thus, we added a section to describe the fitness or application with yellow color in the introduction in the revised word file: “In various development, wideband system is still rapidly expanding, since the unlicensed use of ultra-wideband (UWB) is authorized from 3.1 to 10.6 GHz the U.S. Federal Communications Commission (FCC) for a variety of applications, for instance, indoor and hand-held systems in 2002, [2]. The proposed direct sequence ultra-wideband (DS-UWB) specifications for wireless personal area networks (WPANs) is further divided into a low band of 3.1-5.1 GHz and a high band of 6.2-9.7 GHz, to avoid the frequency use of IEEE 802.11a wireless local area networks (WLANs) at 5-6 GH.”

We have also added a description in the end of the experiment section with yellow mark: The filter example is designed at 4.2 GHz with a 3-dB fractional bandwidth of 55 %, which can meet the low band of 3.1-5.1 GHz of the DS-UWB specification with a slight tuning of the passband.

Reviewer 2 Report

 This paper describe the study and implementations of a Miniaturized Wide-band Band-pass filter.

 The abstract is very good, mentioning all the important topics described on the paper.

 In line 25 (key words) the filter should be band pass filter.

 In the introduction, the authors should refer in what RF application this filter can be used, such as WiFi, 4G ,5G  or other standards. The authors should compare their work with more recent research in this field, and be more explicit in the main innovation presented in this paper.

 In figure 1 the Variable L1/W1, L2/W2 L3/W3, L4/W4 and t are not described in the following text.  It should be before line 87.

 In line 104 the authors start to explain the current distribution. The presented results should be more detailed explained and how is the method used to obtain these results.  In figure 4, the scale should start at 0dB and not -0dB.

 Line 127: the calibration method should be described.

 In line 141, the authors mentioned that : "...proposed filter design is very useful for the modern wide-band wireless communication system...." Should be mentioned what is the main scope, e.g., the standard used in this frequency range. 

 The obtained conclusions are correct, however should be mentioned future work.

 The references date should be more recent. Form 13 references, only 3 has less than 10 years... the authors should consider gather more recent works and compare to the obtained results with more recent papers.

Author Response

The comments from the reviewer are very constructive to our paper. The paper is revised in accordance with the reviewer’s comments and has marked the corrections using yellow color in the revised word file. The attached are the revision and the reply letter for the comments. There are listed as follows:

Answers for Reviewer 2's Comments

This paper describe the study and implementations of a Miniaturized Wide-band Band-pass filter.

 The abstract is very good, mentioning all the important topics described on the paper.

Q1: In line 25 (key words) the filter should be band pass filter.

Reply:

The comment is well received. We have marked this correction using yellow color in the revised word file.

Q2: In the introduction, the authors should refer in what RF application this filter can be used, such as WiFi, 4G ,5G  or other standards. The authors should compare their work with more recent research in this field, and be more explicit in the main innovation presented in this paper.

Reply:

The comment is well received. We added a section to describe the fitness or application with yellow color in the introduction in the revised word file, as following: “In various development, wideband system is still rapidly expanding, since the unlicensed use of ultra-wideband (UWB) is authorized from 3.1 to 10.6 GHz the U.S. Federal Communications Commission (FCC) for a variety of applications, for instance, indoor and hand-held systems in 2002, [2]. The proposed direct sequence ultra-wideband (DS-UWB) specifications for wireless personal area networks (WPANs) is further divided into a low band of 3.1-5.1 GHz and a high band of 6.2-9.7 GHz, to avoid the frequency use of IEEE 802.11a wireless local area networks (WLANs) at 5-6 GH.”

Since this design uses only two quarter wavelength stepped-impedance resonators (SIRs), the filter structure is very simple and the filter size is miniaturized. A wideband bandpass response is obtained and the bandwidth can be tuned by controlling the feeding position of the input/output ports. Moreover, in this design, with using two SIRs, the bandwidth can be further extended and the third harmonic of the center frequency is suppressed.

The main innovation presented in this paper is highlighted to be more explicit in this revised paper.

Q3: In figure 1 the Variable L1/W1, L2/W2 L3/W3, L4/W4 and t are not described in the following text.  It should be before line 87.

Reply:

The comment is well received. We have added a description with yellow mark to define all variables of the Figure 1 before line 87, as followed: (L1, L2) and (W1, W2) are physical lengths and widths of the high impedance section and low impedance section of the SIR 1, respectively. (L3, L4) and (W3, W4) are physical lengths and widths of the high impedance section and low impedance section of the SIR 2, respectively. Gap (g) is the spacing between the SIR 1 and SIR 2. Expression (p) is physical length from the input/output ports to the short ends of SIR 1 and SIR 2.

Q4:  In line 104 the authors start to explain the current distribution. The presented results should be more detailed explained and how is the method used to obtain these results.  In figure 4, the scale should start at 0dB and not -0dB.

Reply:

The comment is well received. We have amended the scale at 0dB. Typically, the current distribution of the filter at the resonant frequency is used to show the location of the maximum and minimum electromagnetic energy. The current distribution at the resonance frequency is used to show where resonance occurs mainly in the structure. This resonant mode can be then excited by providing the suitable input and output terminals in the resonant excitation location. Instead, the resonant mode can be suppressed if a dispersing device is used in the resonant excitation location to avoid the resonant mode. In this revised version, we have presented the current distribution more detailed in Figure 4 and Figure 6 of the revised version.

Q5: Line 127: the calibration method should be described.

Reply: The comment is well received. Before measurement, two coaxial cables of the network analyzer which will connect to the I/O ports of the fabricated filter sample is carefully calibrated by using short-through-open-load calibration steps are carefully processed to make sure that the S21 is close to zero when the two coaxial cables are connected with through device. We have added this description with yellow mark in the revised version.

Q6: In line 141, the authors mentioned that : "...proposed filter design is very useful for the modern wide-band wireless communication system...." Should be mentioned what is the main scope, e.g., the standard used in this frequency range. 

Reply:

The comment is well received. In this revised version, we have re-designed and re-fabricated the new filter with a passband from 3.3-5.8 GHz to meet the useful communication system. We have added a description in the introduction section with yellow mark: The unlicensed use of ultra-wideband (UWB) is authorized from 3.1 to 10.6 GHz the U.S. Federal Communications Commission (FCC) for a variety of applications, for instance, indoor and hand-held systems in 2002, [2]. The direct sequence ultra-wideband (DS-UWB) specifications for wireless personal area networks (WPANs) is further divided into a low band of 3.1-5.1 GHz and a high band of 6.2-9.7 GHz, to avoid the frequency use of IEEE 802.11a wireless local area networks (WLANs) at 5-6 GH.

We have also added a description in the end of the experiment section with yellow mark: The filter example is designed at 4.2 GHz with a 3-dB fractional bandwidth of 55 %, which can meet the low band of 3.1-5.1 GHz of the DS-UWB specification with a slight tuning of the passband.

Q7: The obtained conclusions are correct, however should be mentioned future work.

Reply:

The comment is well received. We have also added a conclusion in the end of the experiment section with yellow mark: Based on this design concept, further works will be processed on the design of dual wideband filter and diplexer with a miniaturized size and a wide stopband.

Q8:  The references date should be more recent. Form 13 references, only 3 has less than 10 years... the authors should consider gather more recent works and compare to the obtained results with more recent papers.

Reply:

The comment is well received. We have added 7 new references, and there are 7 references which their published years are less than 3 years.

We have also added a table to compare our work to these works, and added a description with yellow mark: Table 1 compares this design to some reported works. The designed filter shows acceptable filter performance when comparing to other filters. In addition, this design shows a simple configuration and a miniaturized size. The filter example is designed at 4.2 GHz with a 3-dB fractional bandwidth of 55 %, which can meet the low band of 3.1-5.1 GHz of the DS-UWB specification.

Table 1. Comparison of filter performances of the proposed filter with the previous published works.

Ref. [11]

Ref.[12]

Ref.[13]

Ref. [14]

Ref. [15]

This work

Center frequency (GHz)

2.3

3

2

2.3

1

4.2

|S11| (dB)

13

11.7

20

>13

15

15

|S21| (dB)

0.35

2.1

0.57

0.35

1

1.2

3-dB FBW (%)

80

107

100

80

123

55

Circuit Size (λg×λg)

0.12 ×

0.22

0.89 × 0.46

No description

0.53 × 0.43

0.17× 0.14

0.3 ×

0.1

Wide stopband

No

No

Yes

No

Yes

Yes

Defected ground

No

Yes

No

No

No

No

Round 2

Reviewer 1 Report

After reviewing the changes and response letter, I feel the work is clear enough in describing the authors' new research progress. I support acceptance of the manuscript.